# Stress state at faults: the influence of rock stiffness contrast, stress orientation, and ratio

**Moritz O. Ziegler[1,2], Robin Seithel[3], Thomas Niederhuber[4], Oliver Heidbach[2,5], Thomas Kohl[4], Birgit Mueller[4], Mojtaba Rajabi[6], Karsten Reiter[7], and Luisa Roeckel[4]**

[1]Technical University Munich, Arcisstraße 21, 80333 Munich, Germany

[2]Helmholtz Centre Potsdam GFZ German Research Centre for Geosciences, Telegrafenberg, 14473 Potsdam, Germany

[3]GHJ – Ingenieurgesellschaft für Geo- und Umwelttechnik mbH & Co. KG, Am Hubengut 4, 76149 Karlsruhe, Germany

[4]Institute of Applied Geosciences, KIT, 76131 Karlsruhe, Germany

[5]Institute for Applied Geosciences, TU Berlin, 10587 Berlin, Germany

[6]School of the Environment, University of Queensland, Saint Lucia, Queensland 4072, Australia

[7]Institute of Applied Geosciences, TU Darmstadt, 64287 Darmstadt, Germany `CE1`

**Correspondence:** Moritz O. Ziegler (moritz.ziegler@tum.de)

**Abstract.** The contemporary crustal stress state is primarily driven by gravitational volume forces and plate tectonics. However, there are various smaller-scale sources such as geological structures and stiffness contrast that perturb stresses and deviate them from the regional pattern. For example, borehole stress analysis in numerous cases has revealed abrupt rotations of horizontal stress orientation of up to 90° when faults are crossed. Herein, we investigate the rotation of principal stress axes at a fault by means of a 2D generic numerical model. We focus on the near field of the fault and the damage zone with a fault parameterized as a rock stiffness contrast. A substantial influence of the far-field stress field in terms of the differential stress and in terms of the stress ratio $R_S = S_1/S\_3$ is shown. Furthermore, the contrast in material properties is the basis for any stress rotation, and in particular the stiffness is demonstrated to have a significant influence. Eventually, the impact of the angle between the fault strike and the orientation of $S_{Hmax}$ is demonstrated. Our results show that the stress rotation is negatively correlated with the ratio of principal far-field stresses. A small angle between the far-field stress orientation and the fault facilitates stress rotation. A high contrast in rock stiffness further increases the stress rotation angle. Faults striking perpendicular to the maximum principal stress orientation experience no rotation at all. However, faults oriented parallel to the maximum principal stress orientation experience either no rotation or a 90° rotation, dependent on the ratio of principal stresses and the rock stiffness contrast. A comparison with observations from various boreholes worldwide shows that in general the findings are in agreement, even though the dip angle proves to have an influence on the stress rotation, in particular for shallow-dipping faults.

## 1 Introduction

The contemporary crustal stress state is a key parameter in the stability and safety assessment of subsurface operations, such as the extraction of raw materials, storage of waste, and exploitation of geothermal energy (Catalli et al., 2013; Müller et al., 2018; van Wees et al., 2018). The most frequently freely available and most easily obtained stress information is the orientation of the maximum horizontal stress component $S_{Hmax}$ which is compiled in the World Stress Map (Heidbach et al., 2018). The $S_{Hmax}$ orientation is often observed to be consistent over large areas and volumes following the general patterns imposed by plate tectonics (Han et al., 2019; Heidbach et al., 2018; Engelder, 1992; Rajabi et al., 2017a; Reiter et al., 2014), but regionally variable patterns are also observed (Konstantinou et al., 2017; Niederhuber et al., 2023; Ziegler et al., 2016c). The $S_{Hmax}$ orientation pattern is influenced by topography of large mountain ranges

(Levi et al., 2019; Reinecker et al., 2010; Zoback, 1992; Engelder, 1992) or sedimentary basin geology (Rajabi et al., 2016a, b; Snee and Zoback, 2018). It has also been hypothesized that the $S_{Hmax}$ orientation is diverted by faults (Dart and Swolfs, 1992; Faulkner et al., 2006; Konstantinovskaya et al., 2012; Li et al., 2023; Schoenball et al., 2018; Yale, 2003). Reiter et al. (2024) showed with a comprehensive study using a generic geomechanical model that large-scale fault-induced stress changes due to a discontinuity beyond a distance of a few hundred metres from the fault core are too small to be resolved by stress data.

Nevertheless, abrupt or gradual rotations of the $S_{Hmax}$ orientation have been observed on a metre scale in boreholes (Barton and Zoback, 1994; Brudy et al., 1997; Massiot et al., 2019; Sahara et al., 2014; Shamir and Zoback, 1992; Wang et al., 2023). Over the last decade, a great increase in high-resolution image logs in various basins provides the opportunity for detailed observation of such small-scale stress rotations in the vicinity of faults at borehole scales (Lin et al., 2010; Rajabi et al., 2022, 2024; Sahara et al., 2014; Talukdar et al., 2022; Zhang et al., 2023). Figure 1a shows an example of stress re-orientation of approximately 90° indicated by borehole breakouts (Bell and Gough, 1979; Heidbach et al., 2018; Engelder, 1992) in the vicinity of (and attributed to) a fault, which has been interpreted in an electrical image log in the Surat Basin of Queensland, Australia (Rajabi et al., 2017a).

The observed change in breakout orientation is the manifestation of a change in the orientation of the maximum circumferential stress at the borehole wall (Fig. 1b), which is a function of the position at the borehole wall with respect to the orientation of the principal stress components $S_1$, $S_2$, and $S_3$ (Kirsch, 1898). The principal stress components result from the transformation of the second-order symmetric stress tensor:

$$\sigma_{ij} = \begin{pmatrix} \sigma_{xx} & \sigma_{yx} & \sigma_{zx} \\ \sigma_{xy} & \sigma_{yy} & \sigma_{zy} \\ \sigma_{xz} & \sigma_{yz} & \sigma_{zz} \end{pmatrix}, \qquad (1)$$

into the main axis system (Eq. 2), where the three remaining components are the three principal stresses that are perpendicular to each other, and it is defined that $S_1 > S_2 > S_3$.

$$\sigma = \begin{pmatrix} S_1 & 0 & 0 \\ 0 & S_2 & 0 \\ 0 & 0 & S_3 \end{pmatrix} \qquad (2)$$

Any changes in the magnitudes of the individual components of the stress tensor $\sigma_{ij}$ are reflected in the principal stress components. In the main axis system, the changes in magnitudes in $\sigma_{ij}$ can also be reflected by changes in principal stress axis orientation (Fig. 1b) However, this only occurs if the deviatoric stress tensor changes (Engelder, 1994). The phenomenon of changes in the magnitudes of the components of $\sigma_{ij}$ that can be observed as changes in the orientation of the principal stress axes is commonly referred to as stress rotation. CE2 It can be observed, e.g. as a change in the orientation of borehole breakouts (Fig. 1a).

Stress rotations are observed at various locations worldwide where boreholes have been drilled through faults while observations of the stress state were possible (e.g. Brudy et al., 1997; Cui et al., 2014; Hickman and Zoback, 2004; Lin et al., 2007). While some faults exhibit a rotation of up to 90° (Lin et al., 2007), others show significantly lower rotation (Cui et al., 2014; Hickman and Zoback, 2004) or considering the uncertainties almost no rotation at all (Yamada and Shibanuma, 2015). Even though stress rotations can be observed in boreholes under some circumstances, knowledge of where to expect stress rotations is important for subsurface operations that specifically target faults for their permeable properties (Freymark et al., 2019; Konrad et al., 2021) or may avoid them for the same reasons but still will encounter them (Gilmore et al., 2022; Long and Ewing, 2004). This raises the question as to which parameters determine a stress rotation and how sensitive they are.

The potential for stress rotation and the magnitude for such rotation are assumed to be determined by the smallest and largest principal stress components, $S_1$ and $S_3$, usually considered in terms of the differential stress $S_1 - S_3$ (Reiter, 2021; Sonder, 1990; Ziegler et al., 2017); the contrast in rock stiffness between the fault and an intact host rock; and the angle of the fault strike represented by a rock stiffness contrast with respect to the orientation of $S_{Hmax}$ (Reiter, 2021; Siler, 2023). Several additional parameters such as rock fabric and geological structures (i.e. faults and fractures) have been suggested as the possible causes of these stress re-orientations at small scales (Faulkner et al., 2006; Hung et al., 2007; Siler, 2023; Yale, 2003).

We investigate the reason for the diverse behaviour of the stress state at faults from a geomechanical perspective and investigate the influence of various parameters on the angle of stress rotation. We focus on the near field of the fault and the damage zone, i.e. the volume around the fault where the rock is significantly fractured and affected in its integrity. We expect a substantial influence of the far-field stress field in terms of the differential stress as shown in previous studies (Reiter, 2021; Sonder, 1990; Ziegler et al., 2017) but also in terms of the stress ratio $R_S = S_1/3$. Furthermore, the contrast in material properties is the basis for any stress rotation and in particular the stiffness (Young's modulus) is demonstrated to have a significant influence (Hergert et al., 2015; Reiter, 2021; Ziegler, 2022). Eventually, the angle between the fault strike and the orientation of $S_{Hmax}$ is investigated as an influential parameter (Reiter et al., 2024; Siler, 2023).

We use a generic geomechanical-numerical model that allows a systematic and comprehensive investigation of the model parameter space in terms of stiffness contrast, stress ratio, and angle of the fault with respect to the far-field stress state. We focus on the near field of the fault at scales of tens of metres and limit the investigation to the linear elastic response of the stress state to rock stiffness contrasts, which has

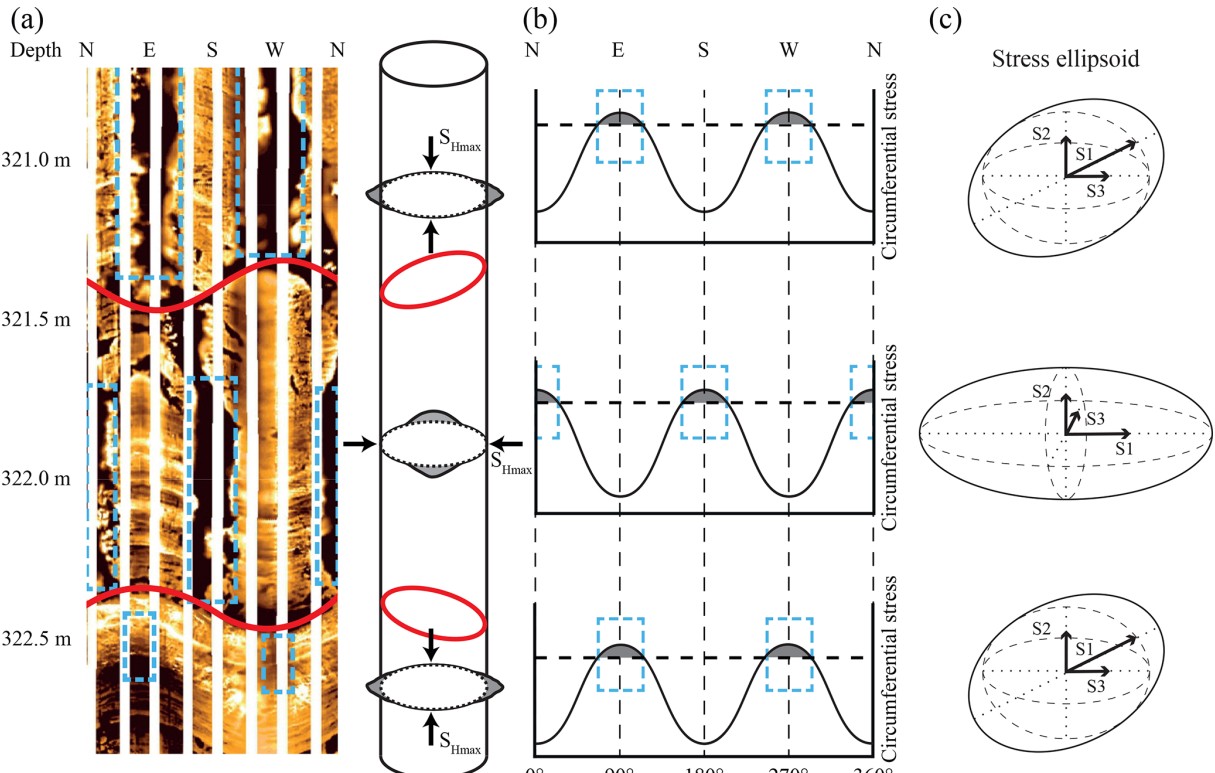

**Figure 1.** An example of stress rotation due to the presence of a small-scale fault in a borehole. The regional $S_{Hmax}$ orientation in the vicinity of this well is approx. 0° (Rajabi et al., 2017a) as indicated by borehole breakouts (black areas in image log) oriented east–west above and below the fault zone delineated by the two sinusoidal red lines. The part in between shows the section that is affected by the fault and thus shows a rotated stress state. **(a)** Borehole breakouts (dashed blue lines) interpreted in a resistivity image log in the Kenya East Well in the Surat Basin, eastern Australia, and the borehole diameter represented in 3D. Dark areas represent conductive zones where the borehole is elongated, while light areas show zones of high resistivity. **(b)** Variations of the circumferential stress at the borehole wall (bold sinusoidal line) with the orientations where the compressive strength (dashed horizontal line) is exceeded (grey areas) and breakouts occur (dashed blue lines). **(c)** Representation of the stress tensor above and below the fault with the stress ellipsoid visualizing orientations and magnitudes of the three principal stresses $S_1$, $S_2$, and $S_3$.

not been investigated in previous studies that focused on processes. Ziegler et al. (2017) did not investigate the full range of parameters (Reiter, 2021) and methods of fault representation due to a different objective (Reiter et al., 2024). CE3
Herein, we conduct an exhaustive study and investigate the entire parameter space.

## 2 Model setup

In order to investigate the stress rotation at a fault dominated by mode-II failure (shear failure) we set up a numerical 2D plane strain model that focuses on the basic characteristics of a material contrast at faults. A generic mature fault zone is assumed with no damage zone outside the core and highly localized damage represented by a 5 m wide zone with a reduction in stiffness. The model assumes a vertical fault in a strike slip stress regime in map view, meaning that the largest principal stress component is $S_1 = S_{Hmax}$ and the smallest principal stress component is $S_2$ due to the limita-

tion to two dimensions and is assumed to be $S_{hmin}$. At the same time, the model is also representative of other types of faults, which requires us to assume the model to be a depth section. For a normal fault $S_1 = S_v$ and $S_2 = S_{hmin}$ or for a thrust fault $S_1 = S_{Hmax}$ and $S_2 = S_v$.

The dimension of the numerical model is $10 \times 10 \, \text{km}^2$ with a central 5 m wide fault that dissects the entire model (Fig. 2a). We use the finite-element method to solve the problem numerically and the model is discretized in a way that boundary effects and numerical artefacts are reduced (Homberg et al., 1997; Spann et al., 1994). Within the fault, plane strain quadrilateral elements form a finite-element mesh with a resolution of 5 m along strike and 0.25 m normal to the fault strike. These elements represent the fault by a rock stiffness that is different to that of the host rock. The 200 m at each side of the fault are also discretized by quadrilateral elements (Fig. 2c). The element size normal to the fault strike increases with distance from the fault. Outside the immediate vicinity of the fault, plane strain triangle ele-

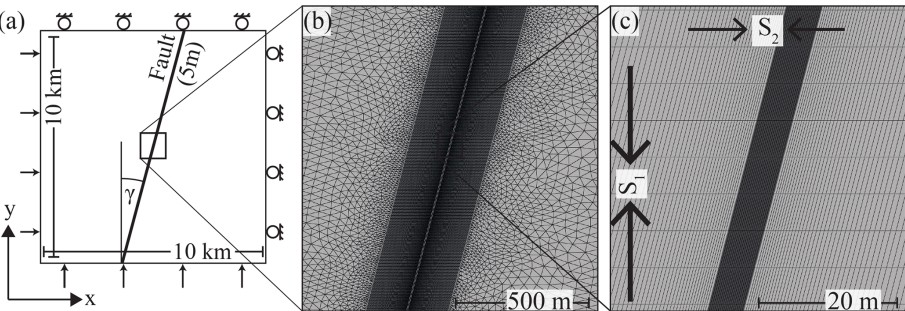

**Figure 2.** Sketch of the model setup. **(a)** Conceptual setup of the generic 2D model with the applied boundary conditions and the location of the fault in the centre of the model. Please note that the fault can be rotated in 5° steps. **(b)** Detailed excerpt that shows the refined mesh geometry in the fault core and close to the fault. An area of 200 m on either side of the fault is discretized with quadrilateral elements for a better resolution. **(c)** Closeup of the fault (dark area) and its immediate vicinity (light area) that shows the high resolution of 25 cm normal to the fault strike and 5 m in fault strike direction. Please note that $S_1 = S_{\mathrm{Hmax}}$ and $S_2 = S_{\mathrm{hmin}}$.

ments with a mesh coarsening to 100 m at the boundary are used (Fig. 2b). In total, 19 realizations of the model were created with different strike angles of the fault. The number of elements slightly varies depending on the strike angle but is always between 145 612 and 172 484 elements. An exception is made for a high-resolution model (732 526 elements) for visualization purposes with a fault strike angle of 15°. The independence of the solution from the discretization is shown in Appendix A. The stress state of the model is controlled with displacement boundaries that are chosen in such a way that they result in the desired horizontal stress magnitudes (Fig. 2a) (Ziegler et al., 2023).

To evaluate which parameter controls the amount of stress rotation, we systematically vary the stiffness contrast between the fault and the host rock, the ratio of principal stresses $R_S = S_1/S_2$, and the angle $\gamma$ between the faults strike and the far-field $S_1$ orientation (Table 1). For the host rock we choose a Young's modulus of $E_{\mathrm{host}} = 40$ GPa and a Poisson's ratio of 0.25, and for the fault the stiffness is systematically varied between $E_{\mathrm{fault}} = 0.4$ and $E_{\mathrm{fault}} = 40$ GPa, resulting in 13 scenarios with a rock stiffness contrast $R_E = E_{\mathrm{fault}}/E_{\mathrm{host}}$ between 0.01 and 1. This corresponds to a range from a very soft fault up to the intact rock. The Poisson's ratio was kept constant at 0.25 as the influence of the stiffness is considered the most important parameter, as shown by previous studies (Reiter, 2021; Ziegler, 2022). We tested the influence of the material contrast in 11 far-field principal stress ratios. We iterated the angle $\gamma$ from 0 to 90° in 5° steps. For the full parameter space this results in total in 2717 model scenarios that were solved and then analysed.

## 3 Results

The rotation of the principal stress axes at a material contrast in a fault centre are investigated. In a sensitivity study, the influence on the rotation of (1) the material contrast ratio, (2) the initial stress ratio, and (3) the orientation between the fault strike and the background stress field orientation are tested. A 2D geomechanical–numerical finite-element model is used to rapidly solve various scenarios that are analysed in the following.

The models provided the normal stress components $\sigma_{xx}$ and $\sigma_{yy}$ and the shear stress component $\sigma_{xy}$ at the nodes of the finite elements. The orientation of $S_1$ in the far-field from the fault is parallel to the $y$ axis (Fig. 2c). In order to obtain the angle of stress rotation we determine the change in the $S_1$ and $S_2$ orientation in the centre of the fault with the initial far-field orientation. The influence of the three individual parameters on the stress rotation angle is investigated in a sensitivity study in the following.

### 3.1 Basic behaviour

The difference between far-field stress state and the altered stress state due to the material contrast are displayed in Fig. 3. In order to initially test the hypothesis that the three parameters, the angle $\gamma$ between the faults strike and the background stress field, the rock stiffness contrast $R_E$, and the relative stress magnitudes of $S_1$ and $S_2$, influence the stress rotation angle, each of the parameters is changed, and the results are observed in the individual panels of Fig. 3 in comparison to Fig. 3a.

The stress rotation for a fault with an angle $\gamma = 15°$ between the strike of the fault and the orientation of the far-field stress component ($S_1$), a moderate stiffness contrast $R_E = 0.4$ (16 GPa fault, 40 GPa host rock), and a principal stress ratio $R_S = 1.4$ ($S_1 = 35$ MPa, $S_2 = 25$ MPa, $S_1 - S_2 = 10$ MPa) is displayed in Fig. 3a. A clockwise rotation of the principal stress axes by 59° can be observed in the fault centre. Outside this zone, the principal stress orientations adhere to the far-field stress state as no visual rotation can be observed (Fig. 3a). This setting is used as a reference and compared to changes due to altered individual parameters in the following. The results are displayed in Fig. 3 and summarized in Table 2.

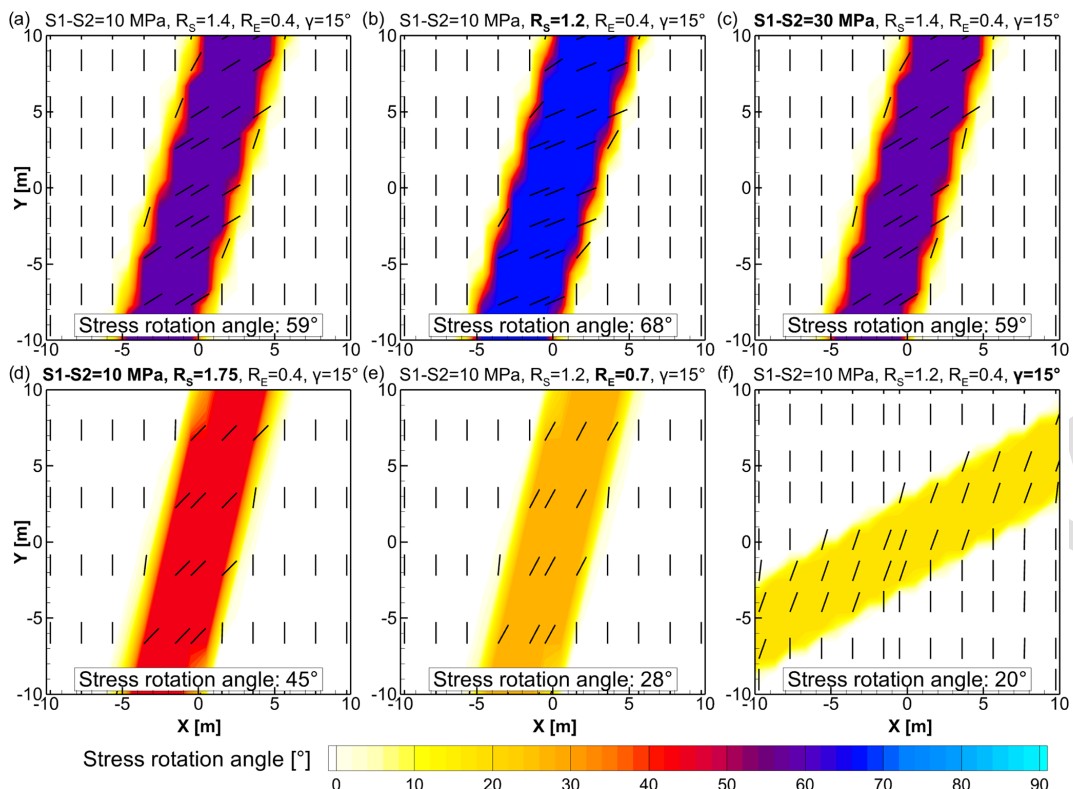

**Figure 3.** Influence of different parameters on the stress rotation angle. The stress rotation is colour coded and shown as vectors or lines that indicate the $S_1$ orientation due to a material contrast in the fault core area (Fig. 2c). **(a)** A basic setting with a fault that is 15° deviated from the orientation of $S_1$, a differential stress of 10 MPa, a stress ratio of $R_S = 1.4$, and a rock stiffness contrast of $R_E = 0.4$ ($E = 16$ GPa in the fault core and $E = 40$ GPa in the host rock). This reference setting is compared with stress rotation based on individually changed parameters (bold in titles of **b–f**). **(b)** $R_S = 1.2$. **(c)** Differential stress increased to 30 MPa and $R_S = 1.2$. **(d)** Differential stress increased to 30 MPa with the same $S_2$ magnitude as in **(a)**, $R_S = 1.75$. **(e)** $R_E = 0.7$. **(f)** Deviation of 45° between fault strike and the $S_1$ orientation. All parameters of all panels are comprehensively listed in Table 2.

A decrease in the stress ratio to $R_S = 1.2$ while the differential stress $S_1 - S_2$ remains constant (but the magnitudes of the principal stresses change to $S_1 = 60$ MPa and $S_2 = 50$ MPa) leads to an increased stress rotation angle of 68°
5 (Fig. 3b). An increase in $S_1 - S_2$ from 10 to 30 MPa ($S_1 = 105$ MPa, $S_2 = 75$ MPa) while $R_S = 1.4$ and therefore constant results in the same rotation angle of 59° as in the reference case (Fig. 3c). In contrast, an increase in $S_1 - S_2$ to 30 MPa ($S_1 = 70$ MPa, $S_2 = 40$ MPa) with a change in stress
10 ratio to $R_S = 1.75$ results in a decrease in stress rotation to 45° (Fig. 3d). This indicates that it is rather the stress ratio $R_S$ that controls the stress rotation than the differential stress $S_1 - S_2$. A more comprehensive comparison is shown in Table 3, which indicates the independence of stress rotation
15 angle from differential stress $S_1 - S_2$ and its dependence on stress ratio $R_S$.

A reduction in the stiffness contrast from $R_E = 0.4$ to $R_E = 0.7$ means that the fault is stiffer and thus closer to the host rock's stiffness. This results in a reduction in the stress
20 rotation of 31 to 28° (Fig. 3e). This positive correlation of rock stiffness contrast $R_E$ to potential stress rotation angle

has been observed previously (Reiter, 2021). It is intuitive in that with the assumptions made herein the same stiffness in fault and host rock ($R_E = 1$) cannot lead to any rotation at all. Thus, greater stiffness contrast (small values for $R_E$) 25 promote less stress rotation.

An increase in the angle $\gamma$ between the fault strike and the far-field $S_1$ orientation from initially 15° to now 45° results in a smaller rotation angle of 19° (Fig. 3f). The angle $\gamma$ is thus negatively correlated with the rotation angle. No rotation is 30 expected for faults that strike perpendicular to the $S_1$ orientation ($\gamma = 90°$). Large stress rotation angles are expected for faults that strike with a low angle towards the $S_1$ orientation.

## 3.2 Spatial effects

As already indicated in Fig. 3, stress rotation is observed 35 only directly in the fault zone where a rock stiffness contrast $R_E$ exists. In order to investigate its influence beyond the fault itself we display the stress rotation angle as a function of distance normal to the fault (Fig. 4). In this idealized representation of a fault without a damage zone, no signifi- 40

**Table 2.** All parameters of the model scenarios tested in Fig. 3 listed according to the panels. CE4

| Panel | $R_S$ | $S_1-S_2$ | $S_1$ | $S_2$ | $R_E$ | $\gamma$ | Rotation |
|---|---|---|---|---|---|---|---|
| a | 1.4 | 10 MPa | 35 MPa | 25 MPa | 0.4 | 15° | 59° |
| b | 1.2 | 10 MPa | 60 MPa | 50 MPa | 0.4 | 15° | 68° |
| c | 1.4 | 30 MPa | 105 MPa | 75 MPa | 0.4 | 15° | 59° |
| d | 1.75 | 30 MPa | 70 MPa | 40 MPa | 0.4 | 15° | 45° |
| e | 1.4 | 10 MPa | 35 MPa | 25 MPa | 0.7 | 15° | 28° |
| f | 1.4 | 10 MPa | 35 MPa | 25 MPa | 0.4 | 45° | 20° |

**Table 3.** Dependence of the stress rotation on the stress ratio $R_S$. The $R_S$ values are compared with different $S_1-S_2$ associated with the same angle $\gamma = 15°$ and the same rock stiffness contrast $R_E = 0.4$.

|  | $R_S = 1.2$ | | |
|---|---|---|---|
| $S_1-S_2$ [MPa] | 6 | 10 | 14 |
| Stress rotation [°] | 67.7 | 67.7 | 67.7 |
|  | $R_S = 1.4$ | | |
| $S_1-S_2$ [MPa] | 12 | 14 | 18 |
| Stress rotation [°] | 59.1 | 59.1 | 59.1 |
|  | $R_S = 1.6$ | | |
| $S_1-S_2$ [MPa] | 18 | 30 | 42 |
| Stress rotation [°] | 50.5 | 50.5 | 50.5 |
|  | $R_S = 1.8$ | | |
| $S_1-S_2$ [MPa] | 24 | 40 | 56 |
| Stress rotation [°] | 43.1 | 43.1 | 43.1 |

cant changes in the orientation of $S_1$ can be observed outside the fault (Fig. 4). However, significantly different stress rotation angles are observed within the fault dependent on the $R_E$ value in line with the observations made in Fig. 3b and e.

Furthermore, within the contrasting rock volumes there is no gradual difference observed between the borders and the centre (Fig. 4). Only at the border between fault and host rock can an intermediate stress rotation angle be observed (transition from grey to white in Fig. 4). This apparent stress rotation should not be interpreted as it is a result of the interpolation from the integration points within the elements to the nodes. At the borders between bodies with different material properties the stress state is a mixture of stress states from the two bounding lithologies and its extent depends on the discretization. It can be disregarded in the following analysis. Thus, in the following we refer to the stress rotation in the centre of the modelled fault only. This allows us to display the results in a more comprehensive way (Fig. 5).

### 3.3 Parameter space

In order to explore the full parameter space, we chose a different visualization. Each stress ratio $R_S$ is represented by

**Table 1.** Range of parameters investigated in this study. Note that the rock stiffness contrast $R_E$ has a step width of 0.1 (4 GPa) between a $R_E = 1$ and $R_E = 0.1$. Larger contrasts are realized to be 0.07, 0.04, and 0.01. The host rock always has a Young's modulus of $E_{host} = 40$ GPa. The stress ratio $R_S$, the differential stress $S_1-S_2$, and the absolute stress magnitudes are indicated. $\gamma$ is the angle between fault strike and $S_1$ orientation that varies between 0° (parallel) and 90° (perpendicular). The results of all computations are provided in the Supplement.

| Parameter | Values | | | | | | | | | | | | | | | | | | | Num. scen. |
|---|---|---|---|---|---|---|---|---|---|---|---|---|---|---|---|---|---|---|---|---|
| $R_E$ | 1 | 0.9 | 0.8 | 0.7 | 0.6 | 0.5 | 0.4 | 0.3 | 0.2 | 0.1 | 0.07 | 0.04 | 0.01 | | | | | | | 13 |
| $E_{fault}$ [GPa] | 40 | 36 | 32 | 28 | 24 | 20 | 16 | 12 | 8 | 4 | 2.8 | 1.6 | 0.4 | | | | | | | |
| $R_S = S_1/S_2$ | 1.0 | 1.1 | 1.2 | 1.3 | 1.4 | 1.5 | 1.6 | 1.7 | 1.8 | 1.9 | 2.0 | | | | | | | | | 11 |
| $S_1-S_2$ [MPa] | 0.00 | 2.25 | 5.00 | 8.25 | 12.00 | 16.25 | 21.00 | 26.25 | 32.00 | 38.25 | 45.00 | | | | | | | | | |
| $S_1$ [MPa] | 20.00 | 24.75 | 30.00 | 35.75 | 42.00 | 48.75 | 56.00 | 63.75 | 72.00 | 80.75 | 90.00 | | | | | | | | | |
| $S_2$ [MPa] | 20.0 | 22.5 | 25.0 | 27.5 | 30.0 | 32.5 | 35.0 | 37.5 | 40.0 | 42.5 | 45.0 | | | | | | | | | |
| $\gamma$ [°] | 0 | 5 | 10 | 15 | 20 | 25 | 30 | 35 | 40 | 45 | 50 | 55 | 60 | 65 | 70 | 75 | 80 | 85 | 90 | 19 |

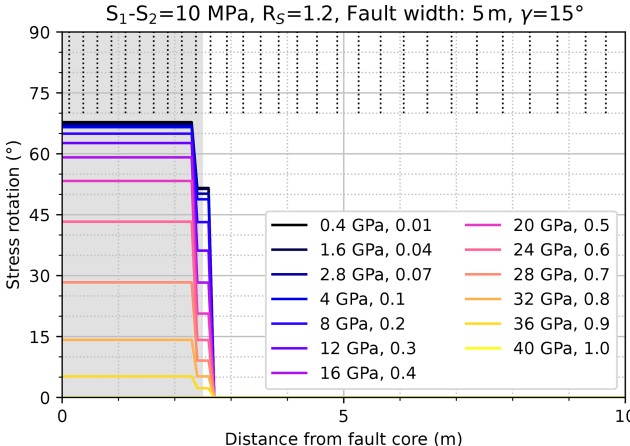

**Figure 4.** Intrinsic stress rotation in the fault zone as a function of increasing rock stiffness contrast. The abrupt change from a rotated to a far-field stress state is observed at the border of the fault. The rotation is shown perpendicular to the fault. The rock stiffness contrast between the fault's Young's modulus and a host rock stiffness of $E_{host} = 40$ GPa is colour coded and indicated in the legend as absolute values in GPa and ratio $R_E$. For reference, the purple line $R_E = 0.4$ is taken from Fig. 3a, while the orange line $R_E = 0.7$ is taken from Fig. 3e. See also Table 2. The distance from the fault core centre ($x$ axis) in relation to the stress rotation ($y$ axis) is shown. The vertical dotted lines at the top indicate the discretization normal to the fault strike with an element size of 25 cm within the fault and increasing outside the fault. Please note that the distance is from the fault core centre, and thus only half of the fault width is shown here in grey.

an individual diagram (panels of Fig. 5). Within each diagram, the angle $\gamma$ between fault strike and $S_1$ orientation is related to the modelled stress rotation angle. Each model scenario with an individual set of parameters is now displayed as a point in a diagram (Fig. 5). The material contrast $R_E$ is colour coded. Four different stress ratios $R_S$ are compared in the four panels of Fig. 5.

The initial observation is the importance of the angle $\gamma$. Structures with a significant contrast in rock stiffness that are perpendicular to the far-field orientation of $S_1$ ($\gamma = 90°$) do not exhibit any rotation of the principal stress axes at all. This is independent of all other parameters. In turn, structures parallel to the $S_1$ orientation ($\gamma = 0°$) can rotate by up to 90°. This signifies a mutual replacement of $S_1$ and $S_2$ orientation. Whether a stress rotation of 90° is reached or not is tied to the ratios $R_E$ and $R_S$.

For large contrasts (i.e. small $R_E$) and small $R_S$ values, the maximum stress rotation angle is reached in several scenarios. At the same time, high $R_E$ values limit the maximally achieved stress rotation angle. The maximum angle for any given $R_E$ value is then up to 90° and controlled by the stress ratio $R_S$ (Fig. 5). However, in these cases the maximum angle is not observed for $\gamma = 0°$ but for angles $0° < \gamma < 45°$.

Apparently, for an angle $\gamma = 0°$ either no stress rotation at all or the maximally possible stress rotation of 90° occurs.

The large influence of the $R_S$ value is additionally displayed in a representation of the three model parameters investigated in this study, where the decisiveness of the $R_S$ value becomes apparent (Fig. 6). This is particularly observable for $R_E = 0.6$ at faults that are striking parallel to the $S_1$ orientation ($\gamma = 0°$). A low $R_S$ of around 1.1 leads to a stress rotation of 90° (Fig. 5a). However, an increase in $R_S$ to $\geq 1.5$ (Fig. 6) prevents any stress rotation at all.

These results show that the previously indicated correlations (Fig. 3) can be confirmed. These correlations are as follows.

- A negative correlation between $R_S$ and the stress rotation angle, i.e. large stress rotations for principal stress magnitudes that are similar to each other.

- A negative correlation between $R_E$ and the stress rotation angle, i.e. large stress rotations for large material contrast (small values of $R_E$).

- A mostly negative correlation between the angle $\gamma$ between strike angle vs. far-field $S_1$ orientation and the stress rotation angle, meaning that $S_1$ that acts normal to a faults strike will see no rotation. An exception is an angle of $\gamma = 0°$, where either no stress rotation or a rotation of 90° is observed.

This shows the combined influence of the stress ratio, the magnitude of material contrast, and the orientation of the fault strike compared to the far-field stress state on the angle of stress rotation. The expected angle of stress rotation can be estimated if all corresponding data are available.

In addition to the previously mentioned correlations, further rules can be observed in Fig. 5. The sum of the angle between fault strike and far-field stress orientation $\gamma$ and the angle of observed stress rotation cannot exceed 90°. No rotation is observed if the fault strikes perpendicular to the far-field stress orientation, i.e. $\gamma = 90°$. A rotation of 90° is only possible if $\gamma = 0°$. However, a rotation of 90° is not observed for all scenarios with $\gamma = 0°$. Whether any rotation is observed or not depends on the stiffness contrast. This highlights the importance of all three parameters on the expected angle of stress rotation.

## 4 Discussion

The influence of the rock stiffness contrast $R_E$, the stress ratio $R_S$, and angle between a fault or a geological structure and the far-field $S_1$ orientation on the rotation of the principal stress axes are derived from a generic study using a 2D plane strain numerical model. The results indicate the importance of the aforementioned parameters on the possibility of occurrence and expected angle of stress rotation. They are

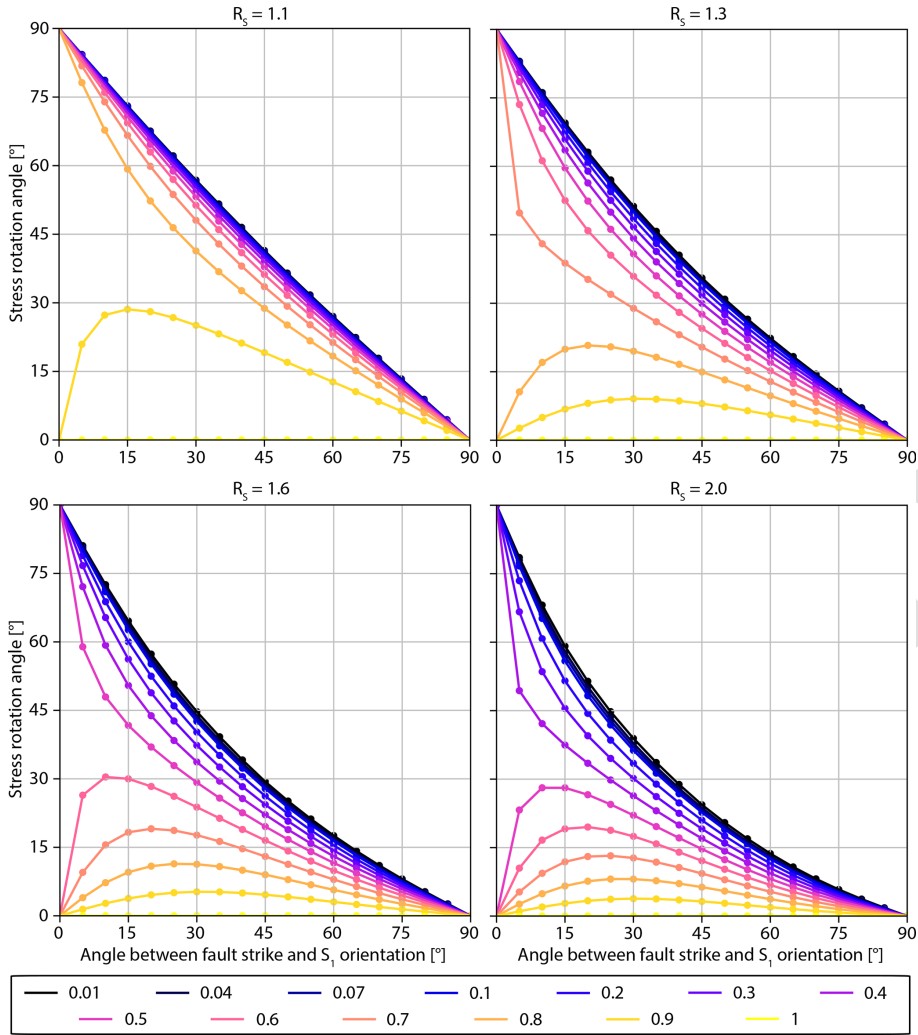

**Figure 5.** The dependence of stress rotation inside the fault core ($y$ axes) on the angle $\gamma$ between the fault strike and the far-field orientation of $S_1$ ($x$ axes), the stiffness contrast $R_E$ between fault core and host rock (colour-coded lines), and the stress ratio $R_S$ (panels). The underlying data are provided in the Supplement.

discussed in light of their general geological implications, impact on geo-energy applications, and limitations.

### 4.1 Implication for fault zones

Faults and fault zones are diverse in their extent, character-⁵istics, and behaviour (Gabrielsen et al., 2017; Childs et al., 2009). While they have several characteristics in common, other features may vary greatly. For example, for some faults a significant damage zone is expected around the fault core that largely depends on the faults maturity (Gabrielsen et al., ¹⁰2017; Childs et al., 2009). Furthermore, the dominant processes that control fault zone behaviour differ, which highlights the complexity of geo-mechanically describing a fault zone.

For the presented study, we assume a weakened fault core ¹⁵compared to the host rock (Casey, 1980; Faulkner et al.,

2006, 2010; Isaacs et al., 2008; Holdsworth, 2004; Collettini et al., 2009). The weakness of fault material is herein implemented as a reduced Young's modulus focused on the fault core assuming a mature fault zone with a clearly delineated core and localized damage. Such a sharp elastic stiff- ²⁰ness contrast between the host rock and the fault core is observed for some fault zones in nature (Barton and Zoback, 1994; Holdsworth et al., 2010). Other (less mature) fault zones have several metres (Barton and Zoback, 1994; Lockner et al., 2009) or even several tens of metres (Faulkner ²⁵et al., 2006, 2010; Li et al., 2012; Williams et al., 2016) of damage zone, which separates the intact rock from the fault core.

To investigate the influence of a significant damage zone as assumed for an immature fault zone, a gradual change in rock ³⁰properties is implemented in an additional model. Therefore, a linear gradient in rock stiffness perpendicular to the fault

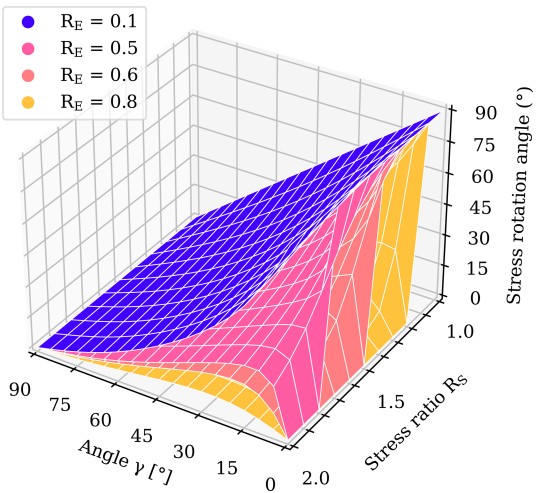

**Figure 6.** A 3D representation of the stress rotation ($z$ axis) and its dependency on the stiffness contrast $R_E$ (colour-coded planes), the angle $\gamma$ between fault and $S_1$ orientation ($x$ axis), and the stress ratio $R_S$ ($y$ axis). To improve visibility, only four different $R_E$ values are displayed.

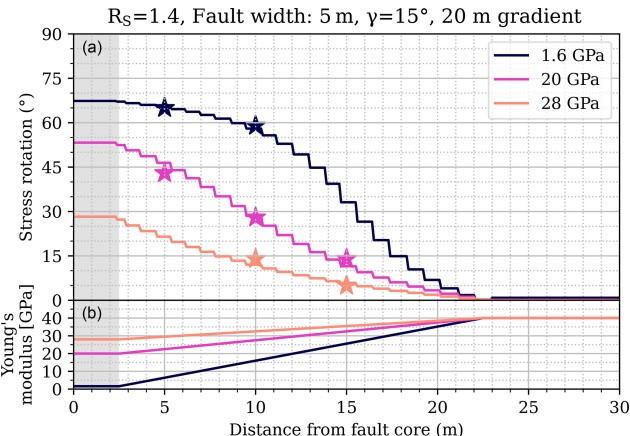

**Figure 7.** Stress rotation in the vicinity of a fault **(a)** with different fault stiffness (colour-coded lines) in contrast to the same host rock stiffness of 40 GPa with a stress ratio $R_S = 1.4$ and an angle $\gamma = 15°$. The damage zone of the fault is realized by an increase in stiffness over 20 m **(b)**. The stress rotation angles expected according to the presented study (Fig. 5) without a gradient of rock stiffness are indicated (stars).

strike increases the Young's modulus from the fault's stiffness to the host rock's stiffness. The resulting stress rotation displays a gradual decrease from the fault core towards the intact host rock (Fig. 7). The angles observed within the damage zone correspond to those observed for the same rock stiffness within the fault core from previous model results (Fig. 5). They do not show any particular deviations from the previously modelled correlations. It is particularly noteworthy that the non-linear dependency on the rock stiffness contrast is reflected in the gradual decrease in stress rotation from the fault core through the damage zone towards the intact host rock.

This shows that the extent of a deviated stress field around a structure with different rock stiffness is dependent on the transition from host rock properties to deviated rock properties. This means that in this simple and generic case the different rock properties in the structure itself do not have any influence on the stress rotation angle beyond its border. Any modelling approach that chooses a representation of faults by a weakened rock material has to include the damage zone – if it is observed and its extent is known – in terms of rock properties.

### 4.2 Application

Modelling of the behaviour of a stiffness contrast with regard to stress rotation is of particular interest for geotechnical ventures. The occurrence of stress rotation is particularly significant for applications such as geothermal energy, where faults are often specifically targeted for their permeability (Barton et al., 1995; Konrad et al., 2021; Seithel et al., 2015; Siler, 2023). At the same time, these faults potentially host

induced seismicity (Gaucher et al., 2015; Schoenball et al., 2014; Seithel et al., 2019). Further geotechnical operations, such as mining and tunnelling, affect the stress state, weaken the rock, and lead to stress rotations themselves (Cai et al., 2022; Ptáček et al., 2015; Ziegler et al., 2015), which can lead to rockfall or other damage. Thus, it is of key interest to estimate the potential for these damage sources, such as (induced) seismicity on pre-existing faults, in advance.

This requires detailed information on the stress state itself, but information on the angle of faults with respect to the stress state is also crucial (Healy and Hicks, 2022; Morris et al., 1996; Röckel et al., 2022; Worum et al., 2004). A regional stress state obtained from geomechanical models often does not include local fault geometries (Ahlers et al., 2022; Clavijo et al., 2024; Gradmann et al., 2024). The modelled stress state is thus then mapped to fault geometries and the according potential for failure is estimated (Röckel et al., 2022; Vadacca et al., 2021; Ziegler et al., 2016a, b). In other instances, faults and their effect on the stress field are included in the model (Hergert et al., 2015; Wees et al., 2003; Xing et al., 2007).

Various different approaches to include faults in a model exist which mimic the different involved processes that have a different impact on the fault behaviour (Henk, 2020; Reiter et al., 2024; Treffeisen and Henk, 2020; Cappa and Rutqvist, 2011). As a result, faults represented in numerical models tend to be implemented in different ways depending on the focus of a study (Henk, 2020; Reiter et al., 2024). Instead of aiming at a realistic representation of a fault, different aspects of a faults properties are commonly investigated individually. This can be the slip behaviour (Hergert et al., 2015; Reiter et al., 2024), the plastic deformation (Nabavi et al., 2018),

or the approach presented in this study as the alteration of rock properties in the immediate fault core and damage zone (Reiter et al., 2024; Treffeisen and Henk, 2020; Cappa and Rutqvist, 2011).

This means that there is a chance that the reactivation potential of a fault is incorrectly estimated if the stress state inside a fault zone is not modelled or is incorrectly modelled. This can happen if the far-field stress state is assumed to be the correct stress state inside a fault instead of a rotated far-field stress state. Röckel et al. (2022) potentially indicate CE5 this challenge, which in some regions could contribute to a mismatch of estimated slip tendency and observed seismicity. Consequently, incorrect estimates of the potential for seismicity could be provided by geomechanical models. Knowledge of the expected stress rotation as a result of the setting provides a first indication of whether significant rotations of the principal stress axes are expected or not.

## 4.3 Observations

Rotations of the $S_{Hmax}$ orientation associated with changes in rock stiffness are mainly observed in boreholes, e.g. by rotating borehole breakouts or drilling-induced tensile fractures (Brudy et al., 1997; Cui et al., 2014; Haimson et al., 2010; Hickman and Zoback, 2004; Lin et al., 2007; Massiot et al., 2019). Some notable examples are shown in Table 4. The observation is inherently subject to uncertainties due to the need for breakouts or drilling-induced tensile fractures to occur in order to be able to identify a rotation. Thus, the number of significant observations of rotations is rather small, and for some projects no rotation at the fault can be observed (Brodsky et al., 2017). In the following we use the investigated correlations to link an observed stress rotation to a stiffness contrast. Therefore, we use available estimates on the stress ratio and observations of a fault in conjunction with the far-field stress orientation.

Massiot et al. (2019) observed significant rotations of the $S_{Hmax}$ orientation in the Taranaki Basin offshore New Zealand. Most prominently a 30° rotation at the intersection of the borehole Whio-1 with the almost vertically dipping Whio-1-fault. In this setting the orientation of $S_{Hmax}$ is approximately parallel to the strike of aforementioned fault (Massiot et al., 2019; Rajabi et al., 2016c). The prevailing stress state is classified as close to transtensional ($S_{Hmax} \geq S_v > S_{hmin}$) with all three stress components inferred. This allows us to estimate a preferential differential stress of 9 MPa, which corresponds to a stress ratio of $R_S = 1.2$, although uncertainties of up to 18 MPa have to be considered in the stress components (Massiot et al., 2019). This information allows us to use the herein-estimated relations to derive a value for the stiffness contrast of $R_E = 0.8$.

Further observations can be made for steeply dipping faults, for example, at the San Andreas Fault Observatory Drilling (SAFOD) site (Hickman and Zoback, 2004) or the Wenchuan fault drilling site (Cui et al., 2014) (Table 4). At the SAFOD site, a differential stress of 64 MPa ($R_S = 2.2$) is expected, with an angle between fault strike and $S_{Hmax}$ orientation of 50° (Hickman and Zoback, 2004). The relations established in this paper predict that the observed stress rotation of approx. 30° requires a high stiffness contrast of $R_E < 0.1$ (Table 4). This is consistent with Zoback et al. (2011) observations in the San Andreas Fault zone.

Stress rotations of up to 20° are observed at the Wenchuan fault drilling project site. A large angle between fault strike and $S_{Hmax}$ orientation and a small stress ratio (Cui et al., 2014) call for a rather moderate rock stiffness contrast of $R_E < 0.7$ in order to explain the observed stress rotation angles (Table 4).

At the German Continental Deep Drilling (KTB) site, 60° stress rotations are observed in unprecedented depths > 7000 m (Brudy et al., 1997). Considering the depth of observed stress rotation, the estimated stress magnitudes are high, as are the differential stress at about 175 MPa and the stress ratio $R_S = 2.4$ (Brudy et al., 1997). The small angle between fault strike and $S_{Hmax}$ orientation allows a 60° rotation to occur even though $R_E \leq 0.3$ is required according to our generic approach.

The Taiwan Chelungpu Drilling Project (TCDP) drilled into the Chelungpu Fault that dips 30° and hosted the 1999 Chi-Chi earthquake (Hung et al., 2007; Lin et al., 2007). Stress rotations of up to 90° have been observed in the vicinity of the fault (Lin et al., 2007). The inferred differential stress and stress ratio, rock stiffness contrast, and observed stress rotation of 90° are well in agreement. However, the angle $\gamma = 90°$ between $S_{Hmax}$ and fault strike should prohibit any stress rotation at all. Interestingly, contrary to the other faults in Table 4, this one is not steeply dipping, which may indicate the relevance of the intermediate principal stress component and the need to investigate the stress rotation in a full 3D modelling approach.

## 4.4 Limitations and outlook

Most of our knowledge on the stress rotations at small scales is from the interpretation of borehole image logs. However, such rotations cannot be observed everywhere, mainly due to the uncertainties in borehole image log interpretation, particularly when it comes to the analysis of borehole breakouts (Azzola et al., 2019; Kingdon et al., 2016). Borehole breakouts are defined as a significant section of the borehole wall that spalls off (Aadnoy and Bell, 1998). In some cases, where breakouts are wide (i.e. opening angles of 15–30°) it is difficult to observe rotation < 30° (Aadnoy and Bell, 1998). While rotation angles > 30° can be observed more easily using breakouts, a higher certainty in observations is expected for drilling-induced tensile fractures, which are usually described as thin (sub)vertical fractures, in vertical boreholes (Aadnoy and Bell, 1998). However, they are less frequently observed and harder to interpret and may be misinterpreted as incipient breakouts in some cases (Rajabi et al., 2017b).

**Table 4.** Observed stress rotations and associated parameters from different (scientific) boreholes that allow for the derivation of the stiffness contrast.

| Location | $S_1 - S_3$ [MPa] | $R_S$ fault strike | $S_{Hmax}$ vs. rotation | Stress angle | Fault dip | Reference | $R_E$ |
|---|---|---|---|---|---|---|---|
| 1. Taranaki Basin | 6 | 1.2 | 0° | 30° | 90° | 1 | 0.8 |
| 2. SAFOD | 64 | 2.2 | 50° | 30° | 70–90° | 2 | < 0.1 |
| 3. Wenchuan Fault | 4–14 | 1.3–1.8 | 69° | < 20° | 70–90° | 3 | < 0.7 |
| 4. KTB | 175 | 2.4 | 0° | 60° | 70–80° | 4 | < 0.3 |

1. Massiot et al. (2019), 2. Hickman and Zoback (2004), 3. Cui et al. (2014), 4. Brudy et al. (1997).

In addition, borehole breakouts and/or drilling-induced tensile fractures need to form in the first place both in the host rock and in the contrasting material in order to be able to detect a rotation at all. Thus, the detection of a stress rotation and (to an even greater extent) the estimation of the material contrast are always subject to uncertainties.

The presented approach in this study investigates elastic rock stiffness contrast in fault zones as the primary reason for stress rotations in a spatially homogeneous far-field stress state. In addition to these static stress rotations, temporal rotations of the principal stress axes can occur co-seismically or post-seismically due to natural seismicity (Hardebeck, 2017; Hardebeck and Okada, 2018; King et al., 1994; Nishiwaki et al., 2018). Further temporal back-and-forth rotations of the principal stress components have been observed (Martínez-Garzón et al., 2013, 2014) and are related to reservoir operations such as injection of fluids (Ziegler et al., 2017) and excavation works such as mining or tunnelling (Eberhardt, 2001; Ziegler et al., 2015). Stress rotation is not only observed at faults. It also occurs if lithological units are mechanically decoupled at a certain horizon due to a contrasting lithology. This also results in a perceived stress rotation in terms of different stress orientations above and below the respective horizon (Cornet and Röckel, 2012; Roth and Fleckenstein, 2001). These changes in rock properties, e.g. in intrusions but also due to faults, can cause massive changes in the $S_{Hmax}$ orientation of up to 90° (Ahlers et al., 2018; Cornet and Röckel, 2012; Rajabi et al., 2024; Reiter, 2021; Tingay et al., 2011). However, the different angles of $S_{Hmax}$ could instead be inherited remnant stresses instead of local deviations due to material contrasts. In other cases stress rotations cannot be attributed to a specific cause (Wang et al., 2023) or anisotropies are assumed (Sahara et al., 2014; Wang et al., 2022). As such, temporal stress rotations and decoupling effects are not covered by the presented work.

The failure of the generically derived relations to predict the expected stress rotation of 90° at the Chelungpu Fault (Table 4) indicates a limitation of the approach. The behaviour of any faults that have a significant deviation from a vertical to subvertical dip apparently do not follow the proposed scheme. This is expected to be a result of its limitation to two dimensions even though the generically modelled fault strike could also represent the dip. That shows the influence of the intermediate principal stress, which cannot be investigated in a 2D model. Furthermore, the observed rotation of the $S_{Hmax}$ orientation is only a part of the actual rotation that likely affects the magnitudes and orientations of all of the principal stress components, as observed by the analysis of focal mechanism solutions, e.g. by Martínez-Garzón et al. (2013, 2014). This indicates the necessity to extend the presented approach to the full 3D stress tensor. In particular, this includes the necessity to investigate the influence of the intermediate principal stress component $S_2$ and its relation to $S_1$ and $S_3$. This should be the aim of future studies as it is well beyond the scope of this work.

## 5 Conclusions

We investigate the relationship between stress rotation in a fault core influenced by (1) a variable rock stiffness contrast compared to the host rock, (2) the stress ratio of the far field, and (3) the far-field stress orientation with regard to the strike of the fault. We use a 2D plane strain model that in a sensitivity study shows that stress rotation is promoted by a low stress ratio and a low angle between the far-field stress orientation and the fault strike. A high stiffness contrast further increases the stress rotation angle. No stress rotation is possible for faults that are striking perpendicular to the maximum principal stress orientation, while faults parallel to the maximum principal stress orientation experience either no stress rotation or a 90° stress rotation, depending on the stress ratio and the rock stiffness contrast within the fault core. The established relations agree with observations from various scientific boreholes worldwide. However, they are only valid for steeply dipping faults where the intermediate principal stress component $S_2$ only has a minor effect. A fully 3D investigation of the established relations including the effect of $S_2$ should be addressed in future research.

## Appendix A: Mesh convergence

The discretization and thus the element size have an effect on model results. In order to prevent interpretation of results

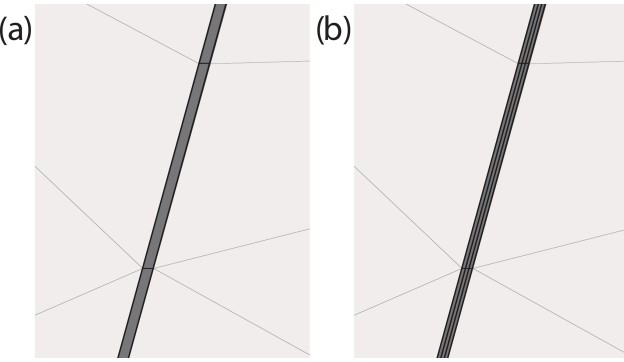

**Figure A1.** Examples of fault discretization. While the absolute number of elements remains approximately the same, the number of elements used for fault discretization varies. **(a)** A single element is used to discretize the fault. **(b)** Three elements are used normal to the fault strike to represent the fault.

influenced by the discretization, a mesh convergence check is performed. The same model geometry is discretized with different element sizes and therefore a different total number of elements. Following this, the different variations are solved and the results are compared, which in this case uses the modelled stress rotation angle.

Herein, two different variations of discretization are investigated. First, the absolute number of elements used to discretize the model. Four different realizations are tested, namely 300, 3000, 170 000, and 700 000 elements. Second, the number of elements that are used to represent the fault normal to the fault strike are varied (Fig. A1). We tested seven different scenarios, namely with 1, 3, 5, 7, 10, 15, and 20 elements normal to the fault strike. All combinations of total numbers of elements and fault elements were combined, meaning that in total 28 different discretization scenarios were tested.

To test the influence of the discretization on the results, the discretizations were compared using an identical set of parameters. The reference set of parameters from Fig. 3a was chosen. This means an angle $\gamma = 15°$ between the strike of the fault and the far-field stress ($S_1$), a stiffness contrast $R_E = 0.4$ (16 GPa fault, 40 GPa host rock), and a principal stress ratio $R_S = 1.4$ ($S_1 = 35$ MPa, $S_2 = 25$ MPa, $S_1-S_2 = 10$ MPa). This set of parameters was used on all 28 different discretization scenarios.

The resulting stress rotation angle that is only dependent on the discretization is shown in Fig. A2. The total number of elements in the tested range does not show any impact on the stress rotation angle. However, the number of elements used for fault representation normal to the fault strike significantly influence the results. This influence, however, is limited to a fault represented by less than three elements normal to the strike (Fig. A2). This effect is observed independent of the total number of elements. These results indicate that the discretization approach realized in this study (20 elements nor-

mal to fault strike, approx. 170 000 elements in total, red dot in Fig. A2) is very well suited to model the stress rotation angle. An influence of the discretization on the results can be ruled out.

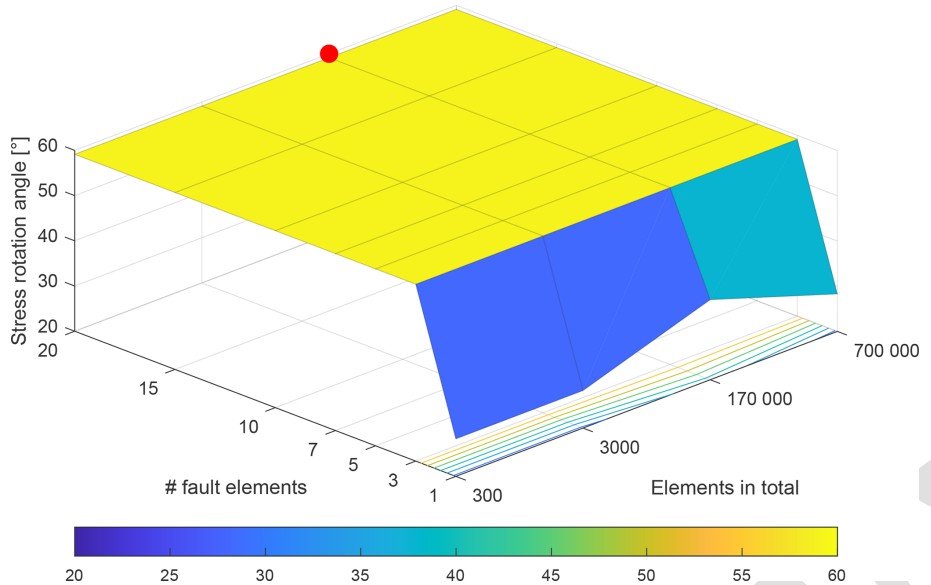

**Figure A2.** Stress rotation in dependence of different discretizations. The total number of elements (*y* axis) and the number of elements used to represent the fault normal to its strike (*x* axis) are put into relation to the modelled stress rotation angle (*z* axis). The approximate mesh resolution used for this study is indicated by the red dot. The tested parameters are the same as in Fig. 3a.

*Data availability.* All modelling results are available in the Supplement.

*Supplement.* The supplement related to this article is available online at: https://doi.org/10.5194/se-15-1-2024-supplement.

*Author contributions.* Conception and study design: RS, TN, MZ, LR, and BM. Software: TN, MZ, and RS. Writing: MZ, OH, and RS. All authors contributed equally to interpretation and discussion.

*Competing interests.* The contact author has declared that none of the authors has any competing interests.

ther geographical representation in this paper. While Copernicus Publications makes every effort to include appropriate place names, the final responsibility lies with the authors.

*Acknowledgements.* The authors express their gratitude to the two anonymous reviewers, whose comments helped to improve the manuscript. The work leading to these results has received funding from the German Research Foundation (DFG grant PHYSALIS 523456847), BGE CE6 SpannEnD 2.0 project, RI CE7 Fabrice Cotton CE8, and the Federal Ministry for the Environment, Nature Conservation, Nuclear Safety and Consumer Protection through project SQuaRe (project number: 02E12062C) CE9. Contributions by Mojtaba Rajabi TS1 were made under his ARC Discovery Early Career Researcher Award (award number DE200101361).

*Financial support.* This research has been supported by the Deutsche Forschungsgemeinschaft (grant no. PHYSALIS 523456847) and the Bundesministerium für Umwelt, Naturschutz, nukleare Sicherheit und Verbraucherschutz (grant no. SQuaRe 02E12062C).

The article processing charges for this open-access publication were covered by the Helmholtz Centre Potsdam – GFZ German Research Centre for Geosciences. TS2

*Review statement.* This paper was edited by David Healy and reviewed by two anonymous referees.

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

**Remarks from the language copy-editor**

CE1 Please note minor edits made to the affiliations for house standard purposes.

CE2 Please confirm the change.

CE3 Please confirm the change.

CE4 Please check this table carefully for changes, as these might not appear in the track-changes file.

CE5 Please check.

CE6 Please define

CE7 Please define.

CE8 Does all of this information belong to the first item (i.e. the DFG grant)?

CE9 Please check this list.

**Remarks from the typesetter**

TS1 Please confirm added name.

TS2 Please note that the funding information has been added to this paper. Please check if it is correct. Please also double-check your acknowledgements to see whether repeated information can be removed or changed accordingly. Thanks.

TS3 Please ensure that any data sets and software codes used in this work are properly cited in the text and included in this reference list. Thereby, please keep our reference style in mind, including creators, titles, publisher/repository, persistent identifier, and publication year. Regarding the publisher/repository, please add "[data set]" or "[code]" to the entry (e.g. Zenodo [code]).

TS4 Please check page range and provide DOI number or URL and last access date.

TS5 Please provide full page range.

TS6 Please provide page range or article number.

TS7 Please provide page range or article number.

TS8 Please check volume and provide page range or article number.

TS9 Please provide page range or article number.

TS10 Please provide page range or article number.

TS11 Please provide page range or article number.

TS12 Please provide volume and page range or article number.

TS13 Please provide page range or article number.

TS14 Please check DOI number.

TS15 Please check page range and provide publisher and DOI number or URL and last access date.

TS16 Please provide volume and page range or article number.

TS17 Please check DOI number.

TS18 Please provide full page range.

TS19 Please check DOI number.

TS20 Please provide page range or article number.

TS21 Please provide page range or article number.

TS22 Please provide journal name, volume and page range (or article number and DOI number) or publisher and DOI number (or URL and last access date.

TS23 Please provide page range or article number.

TS24 Please provide page range or article number.

TS25 Please provide page range or article number.

TS26 Please check initials of last author.

TS27 Please check URL and provide last access date.

TS28 Please provide volume and page range or article number.

TS29 Please provide page range or article number.

TS30 Please provide page range or article number.

TS31 Please provide volume and page range or article number.