# Peer review of "Stress state at faults: The influence of rock stiffness contrast, stress orientation, and ratio"

_EGUsphere, 2024_

## Author Response (AR1)

Dear Editors of Solid Earth, Dear Dr. Healy,

We appreciate the work of the two anonymous reviewers which is highly valuable and helped to improve the manuscript. In the following, we address the issues raised by the reviewers point-by-point and explain our according actions. Our reply closely follows the public discussion on Solid Earth Discussion. The changes that were implemented following the discussion are highlighted in the manuscript file titled "tracked changes".

One issue that has been addressed by both reviewers is the title of the manuscript. Our suggestion for an updated title is "Stress state at faults: The influence of rock stiffness contrast, stress orientation, and ratio". This incorporates comments made by both reviewers, that the stress ratio and far-field stress state are required to be included in the title.

best regards,

Moritz Ziegler

Reviewer #1

We thank the reviewer for the detailed and thorough comments that are very helpful to improve and clarify the manuscript. In our replies, we follow the report structure of the reviewer. First, we address the major points that were raised. Individual and specific remarks will be addressed later in the reply.

Results: We agree with the reviewer, that an introduction to the question that is to be answered is beneficial for the results section and was missing from the initial submission. To avoid to many repetitions, the introduction remained rather brief in the revised version of the manuscript. However, we updated the results section and added subsections to clarify the individual statements and help guide the reader. The visualization has been explained more in detail at corresponding locations in the text. The figure captions have been improved as well.

Instead of the suggested explanation of the visualization strategy in the beginning (or introduction section of the Results section), we preferred to subdivide the Results section into subsections. This allows to guide the reader from the easily understandable Figure 3 (visual representation of the results) to the advanced Figure 4 (which shows that the results can indeed be represented by a single location in the centre of the fault core) to the final results displayed in the meta-visualization of Figure 5. Furthermore, we ensured during typesetting (and will do so during proofing stage) that the figures are closer to the corresponding text which will also help in understanding.

The figure captions were improved. In particular with respect to Figure 4, it is highlighted in the text and caption that the abrupt change in stress rotation angle at the border of the fault is a result of the abrupt change in rock properties typically associated with a mature fault zone. According references with respect to the relation of such an idealized fault zone to geological situations are included.

Discussion: We thank the reviewer for pointing out the weakness in the Discussion section. In particular, the names of the Subsections (and their order) was confusing (see also comments and reply to reviewer #2). We re-structured and re-wrote the Discussion section to include only four subsections that address:

- Implications of the results for fault zones and the geological context.
- Implications of the results for geotechnical applications.
- Observation of stress rotation at individual (scientific) boreholes through fault (zones).
- Limitations of the work and a short Outlook.

This new structure avoids repetitions and puts the work more into perspective than the previous discussion section.

Figures: We agree with the reviewer that the figures and captions can be improved and thank for the ideas and suggestions. We took actions as to improve – where necessary – the figures and the captions. Individually, we have to disagree with the suggestion to decolourize artifacts. They are a part of the model results, and their existence needs to be taken into account. A mentioning of the features being artifacts in the text and figure caption – in our opinion – should be sufficient to advise readers on this issue.

Language: The manuscript was read by a colleague with proficient English language knowledge.

| Reviewer #1 comments | Our actions |
| --- | --- |
| Line 1: I think "body forces" are better than "gravitational volume forces". Perhaps "contemporary stress" and associated concepts as defined by e.g. Engelder (1983) is preferrable (Engelder,T.,1993: Stress Regimes in the Lithosphere. Princeton University Press, 467pp.) | We disagree with the reviewer to replace "gravitational volume forces" with "body forces". The former term is clearly more specific and unambiguous while the latter can be misinterpreted by readers unfamiliar with the subject.

We agree that it should be noted that the contemporary or current in-situ stress state is referred to. We changed the abstract accordingly. |
| Lines 2-3: "when faults are crossed ……". Suggestion: "along or in the vicinity of faults ….. control the amount of principal stress axes." | We assume the reviewer means "controls the amount of principal stress axes *rotation*" and agree with that and changed it accordingly. |
| Line 4: This is fine and is where the important information really starts: "We investigate…." | We agree with the reviewer and accordingly altered the abstract. |
| Line 7: "General findings…." You mean findings in the present work. Please clarify. | We agree with the reviewer and changed the according phrasing. |
| **Introduction & Model setup** | |
| Line 15: parameter **in** the stability assessment | Agreed and changed. |
| Line 16: "exploitation" rather than "usage"?  Add safety assessment? | Agreed and changed. |

| Reviewer #1 comments | Our actions |
|---|---|
| Line 17: Most frequently? You mean: "Information about $S_{Hmax}$ is most easily obtained from…"? | Actually, we mean "most frequently" in that most (publicly) available data is on stress orientations rather than magnitudes. However, we absolutely agree with the reviewer that it is also often most easily obtaine. The text is expanded to include both. |
| Lines 19-24: You mean far-field stress, locally imposed by stress imposed by local……. like topography. Perhaps an advanced textbook (e.g. Engelder 1983) with a summary of such stress systems should be referred to and nomenclature and definitions adapted to that source (see also above). | References to the mentioned textbook is included here and in other appropriate locations in the text. |
| e.g. Lines 23-28 and generally: The authors have a tendency to over-use adverbs in beginnings of phrases: "Furthermore, it has……", "However, Reiter ….", "Nevertheless, on a meters scale….". These examples are taken from sentences that follow each other. | We thank the reviewer for pointing this out and reduced the number of adverbs. |
| Line 29: "across the world" is a surplice phrase: (Where else?) | Agreed. We removed the phrase. |
| Line 33: Borehole breakouts: Again, I think it would be convenient to refer to an advanced textbook – Again, I think Engelder (1983) can be used. | Agreed. According references were added. |
| Line 47: Perhaps this section should start with reference to empiric data/observations: Examples are given in lines 58-60, so I suggest that this is moved here. | We agree with the reviewer and moved and slightly rephrased the according passage. |
| Lines 43-49: Not well phrased. Please separate int two statements and clarify. | We agree that a rephrasing was necessary as this is one of the key points that need to be understood by a reader. We rephrased and expanded accordingly. |
| Line 50: Rather: "This raises questions **as to** which parameters **determine**…." | Agreed and revised. |
| Lines52-53: Rephrase. Suggestion: (if this is what the authors mean to say. If I misread this, a full rephrasing is necessary). "The potential for stress rotation and the magnitude for such rotation are assumed to be determined by…. , the contrast in rock stiffness between ….. and the angle between $S_{Hmax}$ and the fault strike". | We thank the reviewer for the suggestion to clarify this phrase and rewrote it accordingly. |
| Lines 61-74: I think this is well phrased. Please include a short version of this in the abstract. | We thank the reviewer for the suggestion and included a shortened and slightly altered version in the abstract. |
| Line 76: I do see the rationale in the application of a plain strain situation in a strike-slip regime. Most faults will be | We agree with the reviewer in general and revised accordingly. |

| Reviewer #1 comments | Our actions |
| --- | --- |
| influenced by shear whatever the overall tectonic regime. Perhaps this generic situation should be explained (to the unprepared reader), and perhaps the term shear" (rather than-strike slip) should be applied (for the same reason). | At the same time, we try to be cautious about the influence of the intermediate stress component which is shown in the Discussion to be likely of importance. We highlighted this issue in the revised Discussion section. |
| Line 77: Why $S_{Hmax}$ and $S_{hmin}$: Misprint or some hidden significance? Either correct or explain. | The capital H in $S_{Hmax}$ and the lower-case h in $S_{hmin}$ are commonly used to further highlight the difference between the two horizontal components including maximum and minimum horizontal stresses (see e.g. the World Stress Map and associated publications). |
| Line 84: Isn't the term tria-elements commonly written with capital T? | We are not sure what is the correct way to write tria-element. A quick search showed that both with a lowercase and a capital T is possible. Thus, we left it like it is. |
| Line 85: In the order of about 75,00 elements?? Either 75,000 elements were used or not? Or did the model define the grid itself? If so, based on what? Whatever: "In the order of about" is a superfluous double description. | We agree with the reviewer that this reads somewhat peculiar. Furthermore, we thank the reviewer for pointing this out, which allowed us to identify a typo. Indeed, the number of elements is higher. Since a variety of models for the variations in strike are setup, the exact number of elements is also variable. Depending on the model set-up, the total number of elements is variable, and it is between 145,612 and 172,484 elements. An exceptionally high number of elements (732,526) was used for a strike of 15° in order to generate high resolution figures (Figure 3).

This issue is clarified in the revised version. Furthermore, the subject gets a larger importance due to the mesh convergence check that is now included as Appendix A resulting from a request from Reviewer #2. |
| Lines 90ff: Why not $E_{host}$ and $E_{fault}$ to bring it in harmony with e.g. $S_{Hmax}$ | It was intended to write subscript host and fault. It is corrected in the revised version. |
| Line 93: "Non-existence of a fault": Why not "intact rock"? When I walk in a street, the street may be empty, which is congruent with "filled with non-existent cars". But we don't say that. | We agree with the reviewer and changed the wording accordingly. |
| Line 94ff: In my opinion care should be taken so that the conclusions gave a consequent grammatical time sence in distinguishing between what was observed in the experiments and what is generally valid for such systems. In other words: We tested the influence...., scenarios were analysed.... . | We agree with the reviewer and changed the wording accordingly. |

| Reviewer #1 comments | Our actions |
|---|---|
| **Results** | |
| Line 98: Please rephrase: this sentence is complex to the extent that is messsage is obscured. Start with the subject "The orientation of ......", refer immediately to Figure 3, and use Figure 2 more actively. | We added a brief introduction to the Results section as suggested by the reviewer in the general remarks. In addition, we added subsections to the Results section to provide a better structure and simplify the readability. Furthermore, we simplified and rewrote parts of the according paragraph. |
| Line 98 -101: Again, why not $s_{xx}$, $s_{yy}$ etc, since s and $_{xx}$ describe different qualities, namely principal stress and vector components? | This was intended but somehow got lost in typesetting. We changed it accordingly. |
| Line 103: Please expand the text somewhat to help the reader a bit here, e.g. "..... an angle (g) of $15°$ between the strike of the fault and the far field stress ($S_1$) a moderate stiffness contrast ($R_E$)...". | Agreed and changed. |
| Line 104: Take out "There," | Agreed and changed. |
| Lines 103-105: I think that the potential of Figure 3 is not fully exploited here, and the reader should be informed to read Figure 3 in concert with Figures 4 and 5. And why is only Figure 3a referred to in the text, completely neglecting b-f? | We agree with the reviewer that a closer interlinkage of Figures 3, 4, and 5 is required. This is mainly done in the following figure captions of Figure 4 and 5.

However, all subpanels of Figure 3 are referred to and discussed with the same detail as Figure 3a in lines 106-120. |
| Furthermore, Figure 3 is rather nitty-gritty and it is mpossible to read all the details. (Particularly 3b is unclearly displayed). I understand that the displayed zonation (e.g.yellow-red-magneta) in e.g. is an artifact (lines 126-130). If so, the color symbols are misleading and the "artifical" color zones should be removed from the figure display and rather commented upon in the figure caption. | We agree that the previous representation of Figure 3 was not ideal. We reworked the figure in a way that is less "nitty-gritty" and easier readable. In particular:
• We zoomed in on the actual fault.
• The distance between the lines indicating the orientation of $S_{Hmax}$ has been increased.
• The font size has been increased.
• In the titles the changes in parameters with respect to panel a) are now highlighted.

Detailed comments on Figure 3 are also made in the beginning of our reply and are quoted in the following: "Individually, we have to disagree with the suggestion to decolourize artifacts. They are a part of the model results and their existence needs to be taken into account. A mentioning in the text and figure caption – in our opinion – should be sufficient to advise readers on this issue." |
| Lines 105-108: I think this information should be given in the very inroduction to the Results-section on a general basis and combined with a description of the strategy | Agreed and revised. |

| Reviewer #1 comments | Our actions |
|---|---|
| for the prsentation of the results (see general comments to the Results-chapter above). | |
| Line 108: You mean: "S1-S2 remains **constant**". Please state clearly whether this implies absolute stress magnitude, or the difference only? | We agree with the reviewer that the absolute stress magnitudes should also be referenced. As a result, we provide them in the text at the according locations. Furthermore, we added a table (Table 3 in the revised version) that lists the parameters and relates them to Figure 3. |
| Line 115: Rather: "Thus, greater $R_E$ promotes less stress rotation" | Agreed and changed. |
| Line 123: Moved to introductory remarks in this chapter? | Due to the restructuring of the Results section, this part has been left at this location and is included in a new Subsection. |
| Line 127-130: This is a strange phrasing and rather confusing.You mean: "The data on stress orientation at the material borders show here are not valid, because.....the apparent stress rotation result from interpolation of **what?......** please rephrase/explain" | The artifacts occur for numerical reasons. The modelled stress state is computed by the numerical solver not at the nodes but at so called "integration points". They are located within the elements. The stress state is then interpolated from the integration points to the nodes that connect elements. The stress state at a node is thus influenced by the stress state from each bordering integration point. In particular, for significant material contrasts, this results in an averaging of the stress states exactly at the border. And that is the artifact which is observed. |
| Also the following sentence ("At material contrasts....") needs to be rewritten e.g.: "At the borders between bodies with different material properties ....". | Agreed and revised. |
| I understand that the apparent zonation (displayed in Figures 3a, b,c,d) are artifacts. This is technical information that should be kept isolated form the valid observations done in the interpretation of the data (perhaps transferred to the introductory section of "Results"). | See above general comments on Figure 3. |
| Lines 130-150: I think I understand what the authors try to state her, but the messages are blurred by overly complex sentences and uclear language:  Pleaese straighten out and avoid over-complex statements like "...(g = $0_o$) may exhibit the maximally possible stress rotation of 90° which signifies a mutual replacement of $S_1$ and $S_2$ orientation, respectively". This is unclear at the best, and should be rephrased. | During the restructuring of the Results section, this paragraph has been reworked and complex sentences have been simplified. |

| Reviewer #1 comments | Our actions |
|---|---|
| Figure 3, figure caption: "... dependent on different setting" meaning what? I assme you mean: "resulting from contrasting sets of model input parameters". Details in this figure are not easily readable (see e.g. 3b). A better explanation is needed and the crucial differnces should be pinted out. E.g. The only difference between a and b is the Rs-value of 1,4 and 1,2. So please axplain what you want to demonstrate. | We agree with the reviewer that the figure caption of Figure 3 needed to be improved. A more detailed explanation of the content of Figure 3 is included in the new subsection 3.1. Also, we made it clearer, which parameters are altered in the individual panels of Figure 3 compared to the reference setting Figure 3a. A Table with all parameters tested in Figure 3 is now included. |
| Figure 4, figure caption: This figure is not well explained. Rather: "Intrinsic stress rotation in the fault zone as a function of increasing ......". The figure caption needs much more attention. Please lead the attention of the reader more clearly to the important relations here, for example the abrupt termination of stress rotation at the fault border. A comment on geological consequences and limitations would be appropriate (see e.g. Gabrielsen et al. 2017 referred to below). | We improved the figure caption by adding several explanations. Furthermore, we highlighted the abrupt change from a rotated stress field to the far-field stress field at the border of the fault or material contrast. |

Reviewer #2

We thank reviewer #2 for the detailed comments and thorough reading of the manuscript. Please find detailed replies to the raised points in the following. We thank the reviewer for finding some typos or minor technical issues.

| Revier #2 comments | Our actions |
|---|---|
| Line 25: Reiter et al. (2024) says stress changes beyond a few hundred meters are too small to see. However, from Fig 4, the rotation is zero as when the distance is >=3m. Could you please explain the order difference (hundred meters in Reiter versus only 3m in Fig 4)? In the current study, it seems the stress changes are indiscernible whenever the distance is >= 3m. | The reviewer is correct in this observation. However, Reiter et al. (2024) state that there is NO rotation in a distance of several hundred of meters from the fault. Given the different scope of the work (Reiter et al: far-field, this work: near-field), the numerical resolution is also very different in that Reiter et al. cannot resolve any stress rotation in the immediate vicinity (meters to 10 meters) of a fault. But this is exactly where this study takes place. Eventually, Reiter et al. mainly refer to the effect of an explicit discontinuity (technically speaking a contact surface with a frictional behaviour) on the stress state. In contrast, in this manuscript a continuous model is applied.

The discrepancy is clarified in the revised version of the manuscript. |

| Revier #2 comments | Our actions |
|---|---|
| Line 77: In the Strike-slip stress regime, shouldn't S1=SHmax and S3=Shmin? I understand the model is in 2D, but I feel it is a bit confusing to say S2=Shmin. Please consider change that. | We agree with the reviewer that this is confusing. However, the choice was made in order to avoid suggesting the modelling has been done in 3D.

A clarification and explanation is added in the revised version of the manuscript. |
| Line 79: Have you conducted a mesh convergence study in FEM? It could be great to have plots (maybe in the supplement) to show the validation of mesh convergence. | We thank the reviewer for the suggestion and added an appendix that shows details on the influence of the discretization.
It indicates that the used discretization/element size is very well suited to investigate the stress rotation and that no numerical impact is to be expected. The main influence is the number of elements that represent the stiffness contrast perpendicular to its strike (in contrast to the overall element sizes). |
| Table 2: The row name should be "S1-S2 [MPa]", instead of "S1-2 [MPa]". | Agreed and revised. |
| Line 113: "A reduction of RE=0.4 to RE=0.7" seems to be an increase, not reduction. I understand that from the value, it is "increase". However, based on the meaning, it could say a reduction of stiffness contrast between fault and the host rock. Please make it clear in the text. | We agree that this is indeed confusing and clarified it in the revised version here and in other corresponding passages in the text. |
| Figure 6: the color-coding looks different as it is in the legend. | Indeed, this occurred during figure preparation due to a shading option that was not disabled properly. It is corrected in the revised version. |
| Section 4.2: section 4.2 could be combined with section 2 model setup. It is all about how you set the model. You didn't do contact modelling, right? | We agree with both reviewers on the criticism of the Discussion and naming of the subsections. Major changes with respect to text and structure have been made in the Discussion section to accommodate the criticism. |
| Section 4.4 and Table 3: The table caption of table 3 shows "parameters in italics are derived". However, the parameters in Table 3 is in italics. In addition, in order to do comparison of well data with the modelling result, intuitively, I will expect authors use Rs, Re, angle from the reference, and have a column called "modeled stress rotation" and compare it with the column "observed stress rotation". However, in the manuscript, it seems authors sometime derived for input parameters, not for modeled stress rotation. Please have more clarifications in the text. | Table 3: Indeed, for some reason, the parameters in italics are not printed in the table correctly. This is changed in the revised version of the manuscript.

Furthermore, we rearranged Table 3 (now Table 4) in order to more clearly show which parameters are derived and which are observed.

In addition, we added some more information in the text on this "case study" which highlights that the approach has to be applied to derive the stiffness contrast |

| Revier #2 comments | Our actions |
|---|---|
| | instead of to derive the stress rotation angle. This is due to the fact that mainly observations of stress rotation angles are available but stiffness contrasts are often not available. |